# Parental Attitudes, Motivators and Barriers Toward Children’s Vaccination in Poland: A Scoping Review

**DOI:** 10.3390/vaccines13010041

**Published:** 2025-01-06

**Authors:** Krystyna Szalast, Grzegorz Józef Nowicki, Mariola Pietrzak, Agnieszka Mastalerz-Migas, Aleksander Biesiada, Elżbieta Grochans, Barbara Ślusarska

**Affiliations:** 1Department of Family and Geriatric Nursing, Faculty of Health Sciences, Medical University of Lublin, Staszica 6 Str., PL-20-081 Lublin, Poland; krystyna.szalast@umlub.pl (K.S.); barbara.slusarska@umlub.pl (B.Ś.); 2Polish Society of Family Medicine, Syrokomli 1 Str., PL-51-141 Wrocław, Poland; mariola.pietrzak@wum.edu.pl (M.P.); agnieszka.mastalerz-migas@umw.edu.pl (A.M.-M.); alek.biesiada@ptmr.info.pl (A.B.); 3Department of Development of Nursing, Social and Medical Sciences, Medical University of Warsaw, Żwirki i Wigury 61 Str., PL-02-091 Warsaw, Poland; 4Department of Family Medicine, Faculty of Medicine, Wroclaw Medical University, Syrokomli 1 Str., PL-51-141 Wrocław, Poland; 5Department of Nursing, Pomeranian Medical University in Szczecin, Żołnierska 48 Str., PL-71-210 Szczecin, Poland; elzbieta.grochans@pum.edu.pl

**Keywords:** vaccination, child vaccination, vaccine refusal, vaccine hesitancy, parent, Poland, scoping review

## Abstract

Background: Vaccination is one of the most effective ways of protecting individuals against serious infectious diseases and their fatal consequences. Objectives: The aim of this scoping review was to synthesize data on parental attitudes toward vaccination and identify factors influencing the motivators and barriers to children’s vaccination based on Polish studies. Methods: The scoping review process and reporting were based on the Preferred Reporting Items for Systematic Reviews and Meta-Analyses extension for Scoping Reviews (PRISMA-ScRs) checklist. In the period between January 2014 and July 2024, the following databases were searched for publications: PubMed, Web of Science, Cochrane, Ebsco, and Scholar Google. Results: A total of 1531 potentially relevant records were reviewed, and 30 original publications from research samples collected in Poland were selected. According to the findings, vaccination rates varied between 100% and 70%, with parental acceptance levels for mandatory vaccination ranging from 99% to 65%. Parents most commonly cited the physician, the nurse, and the Internet as their primary sources of vaccination-related information. Moreover, parental primary motivators for vaccinating their children were prevention against infectious diseases, the opinion that vaccines are safe, and the belief that childhood vaccination is right and effective. The major barriers to vaccination were fear of vaccine side effects and the belief that vaccines are ineffective. Parents that were better educated, were of younger age, lived in cities, and had a higher income were much more likely to vaccinate their children. Conclusions: Understanding parental attitudes toward vaccination may help develop an educational program aimed at combating misinformation and increasing childhood vaccination coverage rates.

## 1. Introduction

Vaccination is one of the most effective ways of protecting individuals against serious infectious diseases and their fatal consequences. Even though there is scientific evidence that vaccines are effective in preventing infectious diseases and disabilities and in saving the lives of millions of children each year, some parents are hesitant and fearful about vaccinating their children [1]. It should be noted that vaccination-related concerns have spread around the world, prompting scientists to investigate the most significant factors influencing parental attitudes toward this issue. There is currently little information about the factors affecting parents’ attitudes and practices toward childhood vaccination and the reasons for refusing to vaccinate their children. According to research that has been carried out, the reasons for hesitancy and barriers to vaccination differ greatly [2] and must be better understood in order to promote childhood vaccination among parents and increase the vaccination coverage rate among children.

In order to maintain herd immunity acquired through vaccination, the percentage of the vaccinated population must be kept at certain levels. It should be underlined that diverse vaccination policies are followed in individual countries. In Poland, in accordance with Polish law, children and adolescents under the age of 19 are required to be vaccinated, and every year, the Chief Sanitary Inspector updates the national vaccination schedule [3]. The Preventive Vaccination Plan for 2024 in Poland includes 11 mandatory vaccinations, including those against tuberculosis, hepatitis B, diphtheria, whooping cough, tetanus, *Haemophilus influenzae* type b, pneumococcus disease, polio, measles, mumps, and rubella. In Poland, mandatory vaccinations are free of charge. Recommended childhood vaccinations are paid for by the patient and serve as a supplement to mandatory vaccinations. They provide greater protection for the child against certain infectious diseases. In Poland, vaccinations are administered as part of primary healthcare. If vaccination is refused for non-medical reasons, parents are subject to a financial penalty [4].

The number of parents who refuse to vaccinate their children has increased globally in recent years [5]. In Poland, for instance, the number of mandatory vaccination refusals has nearly doubled over the past five years, rising from 48,600 in 2019 to 87,300 in 2023 [6]. Indeed, vaccine hesitancy is a growing global health threat that is limiting immunization and disease prevention efforts. The results show that vaccine hesitancy is highly prevalent and varies globally, ranging from 10 to 40% [7,8].

Vaccine hesitancy is defined as a delay in acceptance or refusal of vaccination despite its availability [9]. In turn, attitudes toward vaccination are understood as an expression of concern or doubt about the value or safety of vaccination [10,11]. Previous research has identified potential reasons for delaying acceptance or refusing vaccination, including the perception of the infection risk being at a low level, decreased confidence in vaccination, and uneasiness about vaccine side effects or its safety [12,13,14]. Vaccine hesitancy results frequently from concerns about the possible risks of vaccination, primarily among parents of children who have to be vaccinated [15]. Parental attitudes toward vaccination are affected by the introduction of new vaccines, reports of side effects, and the spread of misinformation from anti-vaccine movements [16]. In addition, parental attitudes may be culturally influenced and vary depending on their socioeconomic status.

There have been numerous qualitative and quantitative studies, systematic reviews, and meta-analyses published on parental attitudes toward childhood vaccinations [17,18] globally, in particular regions of the world [19], and lastly, in specific countries [20]. Other systematic reviews and meta-analyses have examined parental attitudes toward vaccination against specific diseases [21,22]. Although a few qualitative studies on parental attitudes toward vaccination have been conducted in Poland, there are no scoping reviews of the aforementioned research. Such a scoping review is required to provide contextual information on parents’ attitudes toward vaccination. It should be stressed that different communities may have different reasons for vaccine hesitancy, such as the allure of conspiracy theories. Therefore, the aim of this scoping review was to synthesize data on parents’ attitudes toward vaccination and identify factors influencing the motivators and barriers to children’s vaccination based on Polish studies.

## 2. Materials and Methods

This scoping review was conducted in accordance with the Preferred Reporting Items for Systematic Reviews and Meta-Analyses extension for Scoping Reviews (PRISMA-ScRs) [23]. The scoping review protocol is not available in any databases and is available from the authors [24]. The PRISMA-ScRs checklist for this study is shown in Appendix A.

### 2.1. Search Strategy and Data Source

In July and August 2024, the PubMed, Web of Science, Cochrane, Ebsco, and Scholar Google databases were searched for all potentially relevant articles. The research team developed a search strategy and determined keywords during discussions. The search strategy is described in Appendix A. In summary, the following keywords were used: vaccinations*, immunization, child* or parents’ attitudes* and its synonyms or antonyms, and Poland*, Polish.

### 2.2. Inclusion and Exclusion Criteria

The inclusion criteria were based upon an assessment of parents’ and/or legal guardians’ attitudes, such as motivators and barriers to vaccinating children, and individual and sociodemographic determinants that were positively associated with vaccination rates among children and adolescents up to the age of 19 (the Preventive Vaccination Plan offers free-of-charge vaccination up to 19 years old). Additionally, the study must have been an original work, accessible in either Polish or English, and published between January 2014 and July 2024. A study was excluded if it did not measure or assess any of the aforementioned variables, or if an article simply expressed opinions and views.

### 2.3. Data Extraction and Quality Assessment

Two authors conducted the initial scoping review independently, using the article titles and abstracts. All relevant studies were exported to Excel, which ensured that duplicates were removed. Each author was then provided with a set number of articles to read in order to determine whether they could be included in the study. In this process, we created a table to arrange all the details, such as the study’s first author, the year of publication, the sample size, the type of participants, their age, gender, study design, sampling method, study site, the child vaccination coverage rate, the parents’ or legal guardians’ acceptance level for mandatory vaccinations, the percentage of people believing in the safety of vaccination, the factors affecting vaccination rates, and the reasons for vaccination. We then combined factors and causes of comparable importance. During the selection and assessment of the study for both the title and/or abstract and the full text, the review authors resolved disagreements through discussion or, if necessary, consultation with a third review author. When relevant, we sought additional information from the study’s authors.

### 2.4. Data Analysis

Parental attitudes regarding childhood vaccinations were categorized, and only factors with a *p* < 0.05 were included. Due to the heterogeneity of the types of outcome variables included in this scoping review, a meta-analysis was not conducted.

### 2.5. Synthesis of Results

Since the identified records were heterogeneous, the characteristics and results presented in the combined tables were summarized using descriptive statistics. A narrative overview of the body of research on factors affecting parents’ or legal guardians’ attitudes and practices toward child vaccination is shown.

### 2.6. Study Quality Assessment

The risk of bias in the included studies was assessed by two independent authors (K.Sz. and G.N.), and discrepancies were settled by consensus with a third author (B.Ś.) for arbitration. In the cross-sectional studies examined, the Agency for Healthcare Research and Quality’s (AHRQ) assessment form was employed [25]. The checklist includes 11 items. Each item is rated as “yes”, “no” or “unclear”, and each item receives one point if the study satisfies the methodological requirements. For “no” or “unclear” ratings, zero points are assigned. Scores between 0 and 3 indicate low-quality research, 4 to 7 indicate moderate-quality research, and 8 to 11 indicate high-quality research.

### 2.7. Ethical Approval

Since we utilized data that had been ethically approved, we did not have to obtain ethical approval for the study in question.

## 3. Results

### 3.1. Selected Articles for Review

The literature search process and the process for selecting studies are detailed in Figure 1. A total of 1531 articles were identified as a result of searches through the following databases: PubMed (115), Web of Science (79), Ebsco (26), Cochrane (4), and Scholar Google (1307). There were 743 duplicates removed from the 1531 articles identified. A total of 621 articles were subsequently qualified in the selection process. After reviewing the titles and abstracts, 563 articles were removed, and the remaining 66 articles were qualified by reviewing the full text. At this point, 36 articles were excluded, and only 30 articles were ultimately selected for inclusion in the scoping review.

### 3.2. Study Characteristics

A summary of the studies selected for our scoping review is presented in Table 1. The total sample size for the 30 studies reviewed is 10,534 parents and legal guardians of children aged 0 to 19. All studies were carried out using a diagnostic survey method with an author’s survey questionnaire. In 25 studies, research material was collected through a paper-and-pen personal interview (PAPI), in 3 articles through a computer-assisted web interview (CAWI), in 1 article through both PAPI and CAWI methods, and in 1 article through computer-assisted telephone interviewing (CATI). The most frequently utilized methods for collecting data were purposeful face-to-face surveys (PAPI), which led to non-probability sampling. A total of seven studies [27,28,29,30,31,32,33] employed random probability sampling, including one with stratified sampling [33]. There were five articles published in 2016 and 2020, four in 2017, three in 2014, 2018, 2019 and 2023, two in 2021, and one in 2015 and 2022. The studies’ sample sizes varied from 96 participants [34] to 2300 participants [35]. The majority of study participants were parents or legal guardians aged between 20 and 40. However, it should be noted that five studies did not clearly specify the parents’ ages [30,36,37,38,39]. Female respondents were more prevalent, with participation rates ranging from 56% [40] to 100% [32,41,42]. Several studies did not clearly specify the gender of the participants [27,29,36,43,44]. The majority of the studies were carried out in local communities, but two studies with a representative sample were conducted nationwide [45,46].

### 3.3. Child Vaccination Rate and Acceptance Level Toward Vaccination

The percentage rates of parents and/or legal guardians who reported that their children had been vaccinated ranged from 100% [34,36,49] for small-scale groups (96–125 respondents) to as low as 70% [53] and 76% [54]. The vaccination rate was greater than 90% in the majority of cases [29,31,32,40,45,48,50,56]. Several studies lacked the aforementioned data [27,33,37,39,43,44,47,52].

Overall, parental acceptance for children’s mandatory vaccination was significantly higher, ranging from 99% [47] to 65% [39], as compared to parental acceptance for recommended vaccination, which ranged between 66.41% [42] and 32.86% [38]. All of the studies reviewed assessed positive parental attitudes toward mandatory vaccination, but only a few examined parental attitudes toward recommended vaccination [29,30,31,32,33,34,35,36,37,39,40,52]. Additionally, it is worth noting that, in the majority of cases, the childhood vaccination rate was higher than the parents’ acceptance level for mandatory vaccination [29,31,32,34,36,40,48,49,50,56]. This may theoretically mean that, in the absence of legally required vaccination for children, the vaccination rate would have been even lower, i.e., consistent with the parents’ acceptance level for mandatory vaccination.

The percentage rate of parents who believed that childhood vaccinations were safe ranged from 38% [37] to 96% [49], with 60–70% being most frequently declared [31,32,33,34,39,41,42,44,48,50,52,54]. The detailed results are presented in Table 2.

### 3.4. Sources of Information on Vaccination

According to the respondents, the most common sources of information about vaccination were physicians, nurses, and the Internet. The frequency of identifying the physician as the source of information ranged from 15% [27] to 96% [34], while the nurse ranged from 15% [27] to 67% [51]. The percentage of respondents who indicated the Internet as the vaccination-related source of information ranged from 14.2% [27] to 88.1% [36]. Other parents (24% to 66%) [29,44,54] and friends [31,34,35,36,51,52] were also significant sources of information (10% to 79.2%) for the parents surveyed. The detailed results are presented in Table 2.

### 3.5. Factors Influencing Parental Decisions to Vaccinate Their Children

Table 2 shows parental motivators, barriers, and sociodemographic and individual determinants related to childhood vaccination. According to 16 out of 27 studies, parents’ primary reasoning for vaccinating their children is protection against infectious diseases [28,30,31,32,34,37,40,41,43,47,48,49,50,51,54,56]. A major factor in parental decisions to have their children vaccinated was the belief that vaccination is safe [32,33,34,39,40,44,51] and right and effective [34,40,44]. Other significant motivators included the obligation to vaccinate children in accordance with the Preventive Vaccination Plan, as well as social and state pressure, threats, child vaccination reminders and recalls, fines, and fear of financial penalties (parents avoiding mandatory vaccinations) [30,34,42]. In addition, the fact that mandatory vaccination is free of charge was a significant motivator for the parents [32,34,43].

The parents’ major barriers to their children’s vaccination were fear of the vaccine side effects [28,29,33,34,35,37,40,41,42,43,44,49,50,51,52,54] or the belief that vaccination was ineffective [29,31,48,50,52,54]. Other parental impediments to childhood vaccination included the belief that children receive too many vaccines in their first years of life [31,33,37,48,49], the opinion that vaccines impede the development of naturally acquired immunity [28,31,41,48,54], or that vaccines contain dangerous and harmful doses of chemicals for children [31,35,49,50,54]. Further influences on children’s non-vaccination included the inability to purchase the recommended vaccines [35,41,49,53,54,56] and the belief that pharmaceutical companies that distribute vaccines are profit-driven [37,51,53].

The following sociodemographic characteristics were found to be significantly positively associated with the parental decision to vaccinate their children: parents having completed secondary or higher education [30,50,52], being of a younger age [41], and being of urban residence [52] and in good financial situation [29,47] Additionally, according to the research, certain individual characteristics of the parents were found to be positively associated with the decision to vaccinate their children. These included: a high level of parental knowledge and awareness about the importance of vaccination [35,37,44], information having been received from the physician about vaccination [35], higher levels of positive parental health behavior [39], and having a high level of satisfaction with information on vaccination provided by medical personnel [47].

**Table 2 vaccines-13-00041-t002:** Parental motivators, barriers, and sociodemographic and individual factors of vaccination rates in developing-age children.

Parental Motivators	Barriers	Socio-Demographic and Individual Factors
Preventing infectious diseases in children [28,30,31,32,34,37,40,41,43,47,48,49,50,51,54,56]	Fear of the vaccine side effects [28,29,33,34,35,37,40,41,42,43,44,49,50,51,52,54]	Parentals having a high level of knowledge about vaccination [35,37,44,45,46]
Fear of the child becoming ill [53]	Vaccine-driven profit for pharmaceutical companies [37,51,53]	Parents being of good financial standing (good financial situation leading to the decision to have the recommended vaccinations administered to the child) [29,47]
Social and state pressure, threats, child vaccination reminders and recalls, fines, fear of financial penalties (parents avoiding mandatory vaccinations) [30,34,42]	Opinion that children receive far too many vaccines during the early years of their lives [31,33,37,48,49]	Parents having completed secondary or higher education [30,46,50,52]
Additional protection against infectious diseases [31,54]	Impeding the development of the child’s immune system [28,31,41,47,54]	Parents having a high level of satisfaction with information on vaccination provided by medical personnel [47]
Preventing hospitalizations for a child [41]	Limiting parental autonomy in child care and vaccination [28,35]	Parents being of a younger-age (parents aged between 25 and 34) [41,45]
Belief that vaccines are safe for a child [32,33,34,39,40,44,51]	Belief that vaccinations are ineffective [29,31,48,50,52,54]	Presence of a male parent [46]
Free-of-charge mandatory vaccinations [32,34,43]	Belief that vaccines contain dangerous doses of chemicals [31,35,49,50,54]	Parents being city dwellers [46,52]
Legal obligation to vaccinate a child [34]	Lack of funds for the purchase of vaccines [35,41,49,53,54,56]	Higher levels of positive parental health behavior [39]
Boosting the child’s immune system [34]	Lack of knowledge about recommended vaccines [53]	Information on vaccination provided by a physician [35,45]
Belief that vaccination is right and is an effective way of disease prevention [34,40,44]	Friends’ opinions and information available on the Internet [32,35,50,52]	Trust in a physician being high [45]
Limitation of access to nurseries and kindergartens for unvaccinated children [42]	Failure of general practitioner to inform parents about recommended vaccinations [53]	
Influx of Ukrainian refugees [38]	Child’s pain, stress, and additional burden resulting from vaccination [41,54]	
	Opinion that the requirement for mandatory vaccination needs to be abolished [31]	

### 3.6. Quality Assessment

Table 3 shows the detailed results of the quality assessment of the studies included in this scoping review. The studies were assessed through the AHRQ checklist. Based upon the AHRQ assessment criteria, 8 studies were rated as low quality, and 16 studies were rated as moderate quality. Out of 30 studies, a total of 6 papers were rated as high quality. According to the AHRQ’s checklist, the differences between high-level and low-level quality research concerned the following items: enumeration of inclusion and exclusion criteria, indication as to whether or not subjects were consecutive, whether evaluator covered up other aspects of the subject, explanation of any patient exclusions from analysis, description of how confounding was assessed, and information on how missing data was handled.

## 4. Discussion

The aim of this scoping review was to synthesize data on parental attitudes toward childhood vaccination and identify determinants influencing parental motivators and barriers to their children’s vaccination. This scoping review only included research conducted in Poland among parents and/or legal guardians. The declining vaccination rates in Poland [6] and other countries [57], as well as the COVID-19 pandemic, have sparked a renewed conversation about vaccinations. The assessment of parental attitudes toward vaccination can help us better understand these issues and plan more effective intervention strategies to increase vaccination coverage rates. Intervention plans for parents of children who are required by law to receive vaccinations should be based on the most commonly reported vaccination-related barriers.

It should be noted that numerous factors impact the vaccination-related decision-making process. One is the provision of solid evidence-based knowledge. Well-prepared healthcare professionals may be a source of such information. As it follows from our scoping review, parents and/or legal guardians most commonly indicated a physician or a nurse as their primary source of information regarding vaccinations.

One of the basic tools available to health professionals for encouraging parents to vaccinate their children is to communicate knowledge about vaccination, its efficacy, and safety [58]. During the visit, a physician examines the patient and responds to all of the inquiries regarding vaccination. Interestingly, nurses ranked second among healthcare professionals providing vaccination-related information. This may result from the fact that nurses usually perform instrumental activities, such as administering injections, which causes them to educate parents less about vaccinations. In addition, Polish law stipulates that a physician must qualify a person for vaccination. From 2023 onwards, however, nurses have become entitled to qualify an individual for the COVID-19 vaccine.

The second significant factor affecting parental attitudes toward vaccination is trust in the healthcare system and healthcare professionals. In fact, the more parents trust their healthcare providers, the more they learn about the benefits and risks of vaccination [59,60]. On the other hand, the literature suggests that a lack of proper communication with healthcare professionals leads parents to seek information on vaccination from other sources, where they frequently encounter false information, unreliable scientific studies, and vaccination myths [61]. Our scoping review has shown that the Internet was the third most frequently indicated source of information on vaccination, followed by friends and other parents.

Access to health information on the Internet is of key importance, as is the ability to search for reliable information, understand it and assess it, and then, use that knowledge to make informed decisions [62]. Social networking sites are also crucial in this field as they are web applications that let users communicate and exchange their ideas. This enables social media users to quickly reach a large audience. With an increasing number of people having access to the Internet and to social networking sites, caregiver attitudes toward vaccination for their children are evolving. Although the primary sources of information about health issues are medical professionals, many parents find that the Internet is a popular and convenient way to find health-related information, including information about vaccinations [63]. As demonstrated in the scoping review by Ashfield et al. [64], parents actively search the Internet for information about vaccinations, including blogs and social networking sites. However, parents face a challenge when dealing with contradictory information, both pro- and anti-vaccination. This results from the fact that anti-vaccination groups are active on social networking sites and post anti-vaccination content, which has the potential to affect public opinion and increase vaccine hesitancy. Access to vaccine-critical social networking sites and blogs has been shown to have a negative impact on vaccine attitudes and increase reluctance to vaccination [65].

The impact of misinformation on parental vaccination decisions may have serious consequences for maintaining herd immunity. Given the variety of sources of health information, when organizing vaccination-related interventions, healthcare professionals and government agencies for public health should collaborate with popular social networking sites to ensure that they offer accurate and reliable information. Such actions were taken by Twitter (prior to its reemergence as X) in the USA, which implemented measures to limit the spread of anti-vaccination content by integrating search tools that direct users to government-approved vaccination websites [66]. In addition, Facebook was utilized to increase HPV vaccination coverage among Danish girls [67].

The analysis of studies included in this scoping review has shown that parents’ and legal guardians’ attitudes toward vaccination are not entirely clear, which is in line with earlier systematic reviews [68] and emphasizes the complexity of parental choice. According to the findings of our scoping review, parents have their children vaccinated for three primary reasons: protection against infectious diseases, belief that vaccines are safe, and belief that vaccinations are right and effective. This seems to be a positive trend.

The above-mentioned parental motivators to have their children vaccinated may stem from the fact that, as previously stated, the majority of information on vaccination is obtained from healthcare professionals. In addition, our scoping review has shown that parents decide to vaccinate their children because of legal requirements and concerns about financial liability. As shown by the results of a systematic review by Lee et al. [69], the introduction of mandatory vaccination increases vaccination coverage rates. Of note, there is a debate about the ethics of mandating vaccination [70], as it may contribute to negative perceptions of vaccinations [71] or increase anti-vaccination sentiments [72]. Pisaniak et al. [73] carried out a cross-sectional study of 2205 mothers coming from nine European countries (Poland, Germany, Slovakia, France, Norway, Serbia, Romania, Greece, and Italy), and their findings show that mothers’ attitudes toward financial penalties for refusal of mandatory vaccination vary by country. The majority of Greek and Polish women believed that such financial penalties should be imposed on parents who refused to have their children vaccinated, whereas the majority of Romanian and French women stated that parents should not be fined. According to the results of the study by Bechini et al. [74], however, while mandatory vaccination and legal penalties for refusal of vaccination are effective strategies for raising vaccination coverage rates, their effectiveness differs between countries due to a variety of factors. Perhaps the WHO’s population vaccination rates could be met through relevant legislation, public education, limitations on access to nurseries and schools, and restrictions on child benefits. 

We have identified a number of barriers that lead to parental vaccine hesitancy. The most frequently reported barriers were fear of vaccination-related side effects, the opinion about the ineffectiveness of vaccination, and the belief that children receive far too many vaccines during the first years of life. The result we obtained is consistent with a previously published systematic review [75]. The above may be attributed to parental lack of trust in vaccination-related information. On the one hand, it is of key importance to provide parents with comprehensive and up-to-date information about the risks and benefits of vaccination. On the other hand, healthcare professionals should foster parental trust in vaccinations by expressing a positive attitude toward them [76]. When designing intervention plans to increase vaccination rates, it is critical not to ignore parental concerns, but to carefully consider them.

Finally, it should be noted that our scoping review found that having had secondary or higher education, being of a younger age, having urban residency, and being in a stable financial situation were the socio-demographic determinants that had the greatest impact on the vaccination-related decision-making process. The majority of these factors are associated with higher socioeconomic status (SES), which is a significant determinant of childhood vaccination rates [77]. Low socioeconomic status, when combined with other contextual factors such as race, ethnicity, sexual orientation, migrant status, and indigenous origin, affects social and health inequalities [78], including childhood vaccination decisions. For instance, communication barriers [79] made it difficult for families in low-SES communities to reach medical facilities in order to have their children vaccinated. The results of our scoping review emphasize the significance of targeting vaccination promotion interventions on low-SES individuals who are more likely to experience health disparities. As a result, vaccination programs and policies should be modified to allow for direct resource allocation to overlooked individuals and communities [80]. However, in order to achieve this goal, health disparities must be monitored both within the country and globally.

There are several limitations to this scoping review. Firstly, this scoping review included studies published in peer-reviewed journals that were searched in PubMed, Web of Science, Cochrane, Ebsco, and Google Scholar databases. It is likely that some studies would meet the inclusion criteria for this scoping review, but have not been published in the databases of the journals mentioned above. Secondly, survey questionnaires, which are based on self-reported data, were employed as measurement tools in the studies reviewed in this paper. As there is no access to vaccination records, it is impossible to confirm the actual vaccination status of children of surveyed parents and/or legal guardians. Thirdly, no studies were excluded due to sample size, despite differences between studies. This resulted from the fact that we attempted to take the most comprehensive approach to this review.

Despite these limitations, our study’s findings confirm previously identified issues concerning parental attitudes, motivators, and barriers toward children’s vaccination. Therefore, we suggest two strategies to address parental hesitancy to have their children vaccinated. The first strategy involves the organization of a nationwide training program for primary healthcare physicians, nurses, and midwives so that they can educate parents about vaccinations and provide them with up-to-date and reliable information. Moreover, healthcare providers could use motivational interviewing to address parental concerns while respecting their attitudes toward childhood vaccination. The second strategy is focused on organizing large-scale vaccination campaigns, particularly on social networking sites while limiting the spread of anti-vaccine content. For the purpose of fostering positive attitudes toward vaccination from an early age, the educational program should not only involve parents but also the children and adolescents who are subject to vaccination. The release of a free booklet with stories about vaccination is one example of a tool used to educate children. The idea is that childhood literature would also encourage parents to reflect on vaccination. An example of the plot might include a character who is unable to be vaccinated due to medical reasons and the story could demonstrate how others can receive vaccination to protect this child.

## 5. Conclusions

The results of this scoping review show a high level of self-declared childhood vaccination in Poland, as well as a high acceptance level for mandatory vaccination. The parents and/or legal guardians surveyed demonstrate a high level of confidence in vaccinations. According to the results of the studies included in this scoping review, the most common sources of information about vaccinations are medical professionals (such as physicians and nurses), as well as the Internet. The major motivators for vaccinating children include the prevention of infectious diseases in children, the belief that vaccines are safe, and the obligation to meet legal requirements to have children vaccinated. The most frequently mentioned barriers are fear of vaccine side effects, the belief that vaccines are ineffective, and the opinion that children receive far too many vaccinations. Therefore, Poland’s health policy should aim to keep and increase vaccination coverage rates by encouraging vaccination and, as the review’s findings show, by maintaining or even tightening legal restrictions, such as making benefits available only for vaccinated children. In addition, a nationwide study utilizing standardized tools is required to examine the attitudes of a representative sample of Poles toward vaccination.

## Figures and Tables

**Figure 1 vaccines-13-00041-f001:**
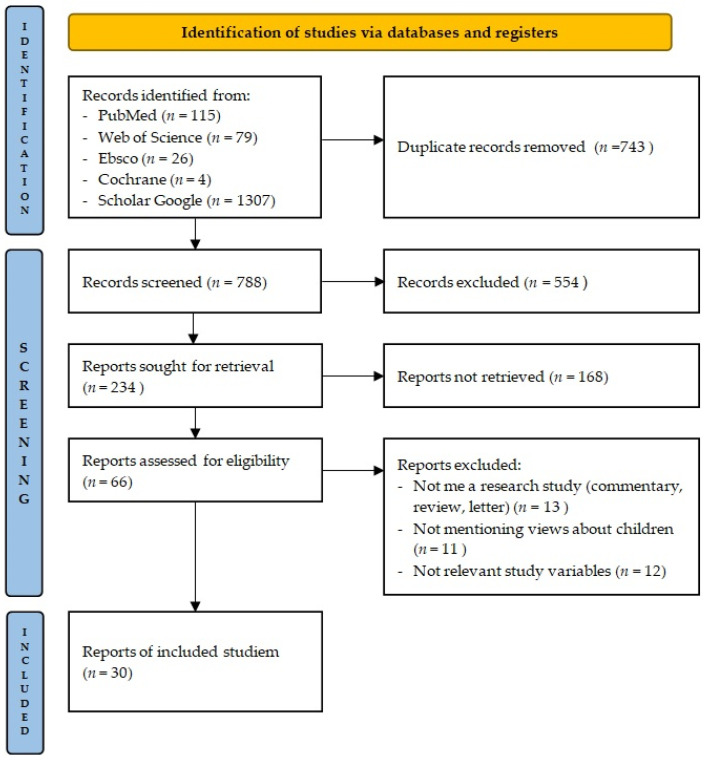
PRISMA flow diagram of study selection [26].

**Table 1 vaccines-13-00041-t001:** Characteristics of the included studies.

Study	Study Design	Participants	Sample Size	Parents’ Age	Parents’ Sex (Female)	Childhood Vaccination Rate	Acceptance Level for Mandatory Vaccination	Acceptance Level for Recommended Vaccination	Vaccine Confidence Level	Sources of Information on Vaccination
Łopata et al., 2014 [36]	Cross-Sectional Study *PAPI*	Parents of children aged 0–6	104	26–35 years old (59%)	N/A	100%	80.2%	N/A	N/A	- physician (75%) - Internet (88.1%), - friends (79.2%).
Cepuch et al., 2014 [47]	Cross-Sectional Study *PAPI*	Parents of children aged 1–5	100	31–40 years old (56%)	66%	N/A	99%	55%	N/A	medical personnel (64%).
Gawlik et al., 2014 [48]	Cross-Sectional Study *PAPI*	Parents of children aged 0–19	123	- 20–30 years old (40.6%), - 31–40 years old (49.6%), - 41–50 years old (9.8%).	91.9%	97.6%	96%	39.8%	61%	- physician (71.5%) - Internet (43.1%), - nurse (43.1%), - TV, radio (16.3%).
Klotško et al., 2015 [37]	Cross-Sectional Study *PAPI*	Parents of children aged 0–19	231	N/A	74.8%	N/A	68%	N/A	38%	N/A
Faleńczyk et al., 2016 [27]	Cross-Sectional Study *PAPI*	Parents of children aged 0–19	104	26–35 years old (59%)	N/A	N/A	98.1%	52.9%	N/A	- information leaflets (15.4%), - physician (15%), - nurse (15%), - Internet (14.2%).
Mrożek-Budzyn et al., 2016 [32]	Cross-Sectional Study *PAPI*	Parents of children right after birth	154	M = 30.1, SD = 5.8	100%	92.2%	90.8%	N/A	61.2%	- physician (78%), - nurse (58%), - TV, radio (34%), - Internet (28%).
Leszczyńska et al., 2016 [34]	Cross-Sectional Study *PAPI* and *CAWI*	Parents of children aged 0–19	96	22–58 years old	92%	100%	83%	43%	76%	- physician (97%), - Internet (27%), - friends (24%), - family (19%), - medical journals (9%).
Trojanowska et al., 2016 [43]	Cross-Sectional Study *PAPI*	Parents of children up to 1 year of age	110	18–40 years old	N/A	N/A	65.5%	45.5%	50.9%	N/A
Pieszka et al., 2016 [49]	Cross-Sectional Study *PAPI*	Parents of children up to 2 years of age	125	- 17–20 years old (1.6%), - 21–29 years old (42.4%), - 30–39 years old (47.2%), - 40–50 years old (8.8%).	90%	100%	98%	46%	96%	N/A
Kotwas et al., 2017 [28]	Cross-Sectional Study *PAPI*	Parents of children aged 0–19	200	- Female: M = 38.41, SD = 5.60 - Male: M = 40.48, SD = 5.37	69.0%	82%	85%	46%	N/A	N/A
Kałucka et al., 2017 [31]	Cross-Sectional Study *PAPI*	Parents of children aged 0–6	170	M = 33.1, SD = 4.2	97%	97.6%	77.1%	N/A	60.9%	- Internet (80%), - friends (65%), - physician (25.3%)
Świątoniowska et al., 2017 [41]	Cross-Sectional Study *PAPI*	Parents of children aged 0–19	102	M = 28.6, SD = 4.3	100%	85.3%	85.3%	60.8%	71.6%	- physician (66.7%), - nurse (9.8%), - midwife (4.9%).
Chudowolska-Kiełkowska et al., 2017 [44]	Cross-Sectional Study *PAPI*	Parents of children up to 2 years of age	120	- ≤20 years old (4.2%), - 20–40 years old (79.2%), - ≥41 years old (16.7%).	N/A	N/A	92.5%	40.6%	75.8%	- medical personnel (37.2%), - Internet (24.2%), - family and friends (23.3%), - TV (5.8%).
Gawlik et al., 2018 [29]	Cross-Sectional Study *PAPI*	Parents of children up to 6 years of age	600	- 20–30 years old (34.2%), - 31–40 years old (54.3%), - 41–50 years old (10%), - 51–60 years old (1.5%).	N/A	98.3%	83.2%	N/A	44.3%	- physician (71.6%), - Internet (55.8%), - nurse (33.3%), - other parents (28.5%).
Braczkowska et al., 2018 [33]	Cross-Sectional Study *PAPI*	Parents or legal guardians of children aged 6–13	1239	22–80 years old	87.4%	N/A	68.2%	N/A	64.3%	N/A
Kraśnicka et al., 2018 [50]	Cross-Sectional Study *PAPI*	Parents of children aged 0–19	300	- 18–30 years old (30%), - 31–40 years old (46.3%), - 41–50 years old (23.7%).	83%	96.3%	90%	53%	62.7%	N/A
Duda et al., 2019 [30]	Cross-Sectional Study *PAPI*	Parents of children aged 0–6	302	N/A	87.7%	88.5%	89.6%	N/A	43.8%	- medical personnel (66.7%), - Internet (8.9%), - medical literature (41.8%).
Kędzierska et al., 2019 [51]	Cross-Sectional Study *CAWI*	Parents of children up to 2 years of age	300	18–43 years old	91%	78.67%	78.67%	38.33%	90%	- physician (87.33%), - nurse (67%), - midwife (28.67%), - Internet (39.33%), - other parents (38.67%).
Stroba-Żelek et al., 2019 [52]	Cross-Sectional Study *PAPI*	Parents or legal guardians of children aged 0–19	233	M = 34.1, SD = 6.57	76.8%	N/A	90%	N/A	63%	- physician (82.4%), - nurse (42%), - Internet (33%), - TV (30.9%), - friends (30.5%), - medical literature (20.6%).
Lewandowska et al., 2020 [35]	Cross-Sectional Study *PAPI*	Parents of children aged 0–19	2300	M = 41.2, SD = 7.01	55%	82%	89%	N/A	N/A	- physician (73%), - nurses (28%), - Internet (15%), - leaflets (12%), - friends (10%).
Kraśnicka et al., 2020 [39]	Cross-Sectional Study *PAPI*	Parents of children aged 0–19	314	- 18–30 years old (30%), - 31–40 years old (46.3%), - 41–50 years old (23.7%).	83%	N/A	65%	N/A	63%	- physician (84%)
Furman et al., 2020 [45]	Cross-Sectional Study *CAWI*	Parents or legal guardians of children aged 0–18	1079	18–88 years old	53.7%	97.7%	N/A	N/A	74.6%	N/A
Janosz et al., 2020 [53]	Cross-Sectional Study *PAPI*	Parents of children up to 3 years of age	100	- <25 years old (28%), - 26–30 years old (37%), - 31–35 years old (25%), - ≥36 years old (10%).	81%	70%	88%	40%	N/A	N/A
Szymoniak et al., 2020 [54]	Cross-Sectional Study *PAPI*	Parents of children right after birth	100	- 26–30 years old (33%).	93%	76%	87%	39%	60%	- medical personnel (81%), - Internet (81%), - other parents (66%), - medical literature (41%).
Pisaniak et al., 2021 [42]	Cross-Sectional Study *PAPI*	Parents of children aged 0–18	646	M = 30.89, 15–65 years old	100%	82.35%	86.68%	66.41%	69.04%	N/A
Sochocka et al., 2021 [55]	Cross-Sectional Study *PAPI*	Parents of children aged 0–18	170	- ≤20 years old (5.3%), - 21–30 years old (23.4%), - 31–40 years old (54.3%), - ≥41 years old (17%).	86.5%	100%	100%	31.42%	N/A	N/A
Łoś-Rycharska et al., 2022 [56]	Cross-Sectional Study *PAPI*	Parents of children aged 0–19	278	- Female: M = 35.98, - Male: M = 38.23.	96.76%	96.4%	93.53%	47.84%	N/A	- physician (88.49%), - posters (36.33%), - leaflets (31.29%), - medical literature (34.5%), - Internet (21.2%).
Cholewik et al., 2023 [38]	Cross-Sectional Study *CAWI*	Parents of children aged 0–19	547	N/A	91.96%	81.9%	90.86%	32.86%	N/A	N/A
Szalast et al., 2023 [40]	Cross-Sectional Study *PAPI*	Parents of children aged 0–19	100	- ≤30 years old (11%), - 31–40 years old (35%), - 41–50 years old (46%), - ≥51 years old (8%).	56%	93%	88%	N/A	53%	N/A
Sobierajski et al., 2023 [46]	Cross-Sectional Study *CATI*	Parents of children aged 9–15	360	28–56 years old	63.1%	68.9%	86.6%	N/A	N/A	N/A

Abbreviations: PAPI: paper-and-pen personal interview; CAWI: computer-assisted web interview; CATI: computer-assisted telephone interviewing; M: mean; SD: standard deviation; N/A: not assessed.

**Table 3 vaccines-13-00041-t003:** Quality assessment results of observational studies (*n* = 30) using the questionnaire according to the Agency for Healthcare Research and Quality (AHRQ).

No.	First Author (Year)	11-Item AHRQ Checklist											Total Score	Quality of Study
		1	2	3	4	5	6	7	8	9	10	11		
1.	Łopata et al., 2014 [36]	Y	Y	Y	N	N	N	N	N	N	Y	N	4	Moderate
2.	Cepuch et al., 2014 [47]	Y	Y	Y	N	N	N	N/A	N	N	Y	N	4	Moderate
3.	Gawlik et al., 2014 [48]	Y	N	Y	N	N	N	N/A	N	N	Y	N	3	Low
4.	Klotško et al., 2015 [37]	Y	Y	Y	Y	N	N	Y	N	Y	Y	N	7	Moderate
5.	Faleńczyk et al., 2016 [27]	Y	Y	Y	Y	N	N	N/A	N	N	Y	N	5	Moderate
6.	Mrożek-Budzyn et al., 2016 [32]	Y	N	Y	N	N	N	N	N	N	Y	N	3	Low
7.	Leszczyńska et al., 2016 [34]	Y	Y	Y	N	N	N	Y	N	N	Y	N	5	Moderate
8.	Trojanowska et al., 2016 [43]	Y	Y	Y	Y	N	N	N	N	N	Y	N	5	Moderate
9.	Pieszka et al., 2016 [49]	Y	N	Y	Y	N	N	N	N	N	Y	N	5	Moderate
10.	Kotwas et al., 2017 [28]	Y	N	Y	Y	N	N/A	Y	N	N	Y	N	5	Moderate
11.	Kałucka et al., 2017 [31]	Y	N	Y	N	N	N	N	N	N	Y	N	3	Low
12.	Świątoniowska et al., 2017 [41]	Y	Y	Y	N	N	N	N	N	N	Y	N	4	Moderate
13.	Chudowolska-Kiełkowska et al., 2017 [44]	Y	Y	Y	N	Y	N	Y	N	N	Y	Y	7	Moderate
14.	Gawlik et al., 2018 [29]	Y	N	Y	N	N	N/A	N	N	Y	Y	N	4	Moderate
15.	Braczkowska et al., 2018 [33]	Y	Y	Y	N	N	Y	Y	N	N	Y	N	6	Moderate
16.	Kraśnicka et al., 2018 [50]	Y	N	Y	N	N	N	N	N	N	Y	N	3	Low
17.	Duda et al., 2019 [30]	Y	Y	N	N	N	N	N/A	N	N	Y	N	3	Low
18.	Kędzierska et al., 2019 [51]	Y	Y	Y	Y	Y	N	N	N	N	Y	N	6	Moderate
19.	Stroba-Żelek et al., 2019 [52]	Y	N	Y	N	N	N	N	N	N	Y	N	3	Low
20.	Lewandowska et al., 2020 [35]	Y	Y	Y	N	Y	N	N/A	Y	Y	Y	Y	8	High
21.	Kraśnicka et al., 2020 [39]	Y	Y	Y	N	Y	Y	N/A	Y	N	Y	Y	8	High
22.	Furman et al., 2020 [45]	Y	Y	Y	N	Y	N	N/A	Y	Y	Y	Y	8	High
23.	Janosz et al., 2020 [53]	Y	N	Y	N	N	N	N	N	N	Y	N	3	Low
24.	Szymoniak et al., 2020 [54]	Y	Y	Y	N	N	N	N/A	N	N	Y	N	4	Moderate
25.	Pisaniak et al., 2021 [42]	Y	Y	Y	N	Y	Y	N/A	Y	N	Y	Y	8	High
26.	Sochocka et al., 2021 [55]	Y	Y	Y	N	Y	Y	N/A	Y	N	Y	Y	8	High
27.	Łoś-Rycharska et al., 2022 [56]	Y	N	Y	Y	N	N	N	N	N	Y	N	4	Moderate
28	Cholewik et al., 2022 [38]	Y	Y	Y	N	N	N	Y	N	N	Y	N	5	Moderate
29.	Szalast et al., 2023 [40]	Y	N	Y	N	N	N	N	N	N	Y	N	3	Low
30.	Sobierajski et al., 2023 [46]	Y	Y	Y	N	Y	Y	Y	Y	N	Y	Y	9	High

Abbreviations: 1: Defines the source of information; 2: Lists inclusion and exclusion criteria; 3: Indicates time period used for identifying patients; 4: Indicates whether or not subjects were consecutive; 5: States whether evaluator covered up other aspects of the subject; 6: Assessments provided for quality assurance purposes; 7: Explains any parental exclusions from analysis; 8: Describes how confounding was assessed; 9: Explains how missing data were handled; 10: Summarizes patient response rates; 11: Clarifies percentage of incomplete; Y: Yes; N: No; N/A: not assessed.

## Data Availability

The datasets used and/or analyzed during the current study are available from the corresponding author upon reasonable request.

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
