# Peer review of "Parental Attitudes, Motivators and Barriers Toward Children’s Vaccination in Poland: A Scoping Review"

_vaccines, 2025, doi:10.3390/vaccines13010041_

Round 1
Reviewer 1 Report
Comments and Suggestions for Authors
Estimated Authors,
I've read with great interest the present Scoping Review on the Parental Attitudes, Motivators and Barriers Towards Children's Vaccination in Poland.
The present study may play a key role in the ongoing struggle against the rising vaccine hesitancy in Eastern Europe, and particularly in Poland, as Authors were able to collect, gather and discuss a large array of moderate or even high quality studies. Unfortunately, due to the heterogenous design and strategies from parent studies, Authors were unable to provide a quantitative summary of the items they recollected, but this issue is clearly far from being a fatal flaw to this study.
Briefly, the key messages from this paper are that (I'm deliberating quoting the abstract):
"Parents most commonly cited the physician, the nurse and the Internet as their primary sources of vaccination-related information. Moreover, parental primary motivators for vaccinating their children were prevention against infection diseases, the opinion that vaccines are safe, and the belief that childhood vaccination is right and effective. The major barriers to vaccination were fear of vaccine side effects and the belief that vaccines are ineffective. Parents that were better educated, were of younger age, lived in cities, and had a higher income were much more likely to vaccinate their children".
These results are obviously well consistent with decades of international research on Vaccine Hesitancy, but is is quite significant to stress how these factors appear as significant and relevant also for the assessed polish settings.
Some minor comments that could be rapidly addressed:
1) Figure 1 is inconsistent with the figures provides across the main text regarding the number of sampled articles (eligibility stage: 66, but the main text provides 38); please double check which estimate is correct;
2) Authors should reconcile their data with being the parent studies only drawn from Poland: are data from Poland possibly overlapping and/or of significance for other European Areas? Which factors from Poland may have inflated and/or deflated recollected data? May have impacted on the reported attitudes towards vaccines the approach to vaccinations during the Socialist regime, with some degree of reaction in the following decades? Moreover: in the main text of introduction, Authors claim that vaccine-hesitant parents may be forced to pay some financial penalties. However, the VH is far from fading to black: according to the knowledge of study Authors, are financial penalties actually delivered and recollected?
Reviewer 2 Report
Comments and Suggestions for Authors
Congratulation for this review that i recommand to be published.
Before final submission, i wonder whether you could give some information or/and comment on the following questions:
- In the section on quality analysis, have you observed significant result differences among the high level quality studies( 20,21,22, 25 and 30) and low level ones?
- Which acceptance or hesitancy factors have you identified for specific diseases which could be prevented from a vaccine?
- Why and to which extend the acceptance level has shown some negative tendancy ( for which vaccine and which population groups)
- From the most relevant findings, what would you advise public health authorities to concentrate strategies and invest in focused interventions in order to enhance acceptance among population groups with weak information or influenced by anti anti vaccination movements
Last, i suggest that your conclusions section be more policy oriented, showing the potential impact of such a review for the future success of vaccination programmes in Poland
Reviewer 3 Report
Comments and Suggestions for Authors
Given the current issues related to vaccination, the authors performed an interesting and quite helpful review. The article is well-structured, and the information is presented clearly. I only recommend a few minor changes:
The authors should restructure the Table 2. They should combine parental motivators, barriers, and socio-demographic and individual factors into a table with three clear columns:
parental motivators,
barriers, and
socio-demographic and individual factors.
The bibliographic references should appear as numerical citations, as follows: Social and state pressure, threats, child vaccination reminders and recalls, fines, and fear of financial penalties (parents avoiding mandatory vaccinations) [30, 34, 42].
I consider that restructuring will enhance readability and provide an organized summary of the findings.
In the Discussion section, the authors should expand on the role of social media in vaccine hesitancy and discuss how social media platforms influence parents' decisions about vaccination, highlighting both the positive and negative impacts of social media.
